# Influence of Maternal Diet and Lactation Time on the Exosomal miRNA Cargo in Breast Milk

**DOI:** 10.3390/foods14061003

**Published:** 2025-03-16

**Authors:** Laura Sanjulián, Alexandre Lamas, Rocío Barreiro, Alberto Cepeda, Cristina Fente, Patricia Regal

**Affiliations:** Food Hygiene, Inspection and Control Laboratory (LHICA-USC), Department of Analytical Chemistry, Nutrition and Bromatology, Faculty of Veterinary Science, Campus Terra, Universidade de Santiago de Compostela (USC), 27002 Lugo, Spain; laura.sanjulian.fernandez@usc.es (L.S.); rocio.barreiro@usc.es (R.B.); alberto.cepeda@usc.es (A.C.); patricia.regal@usc.es (P.R.)

**Keywords:** breast milk, miRNA, diet, Mediterranean diet, Southern European Atlantic diet, bacteria, fatty acids, minerals, Spain, qPCR

## Abstract

The importance of breastfeeding for maternal and infant health is widely accepted. In this sense, the presence of macro and micronutrients in human milk ensures proper and safe infant nutrition, along with multiple bioactive molecules that modulate the newborn’s immunity. Usually packaged within exosomes, miRNAs circulating in milk are bioavailable to breastfed infants. Their role in infant development is poorly understood, mainly because the miRNA cargo of human milk has not been fully elucidated to date. The objective of this study is to assess the presence of eleven miRNAs (miR-148a-3p, miR-29b-3p, miR-125b-5p, miR-200c-3p, let-7f-5p, let-7b-5p, let-7a-5p, miR-92a, miR-181a-3p, miR-30a-5p and miR-155-5p) in breast milk exosomes and determine the impact of lactation time and maternal factors on their levels. Samples were collected from a cohort of 59 Spanish mothers, ranging from 1 to 59 months of lactation. MiRNAs were analyzed using reverse transcription and qPCR. Lactation time showed a significant and noteworthy downregulation for miR-148a-3p, miR-200c-3p and miR-125b-5p and hsa-let-7a-5p. The levels of miRNAs were correlated with the levels of some fatty acids such as arachidonic acid and DHA. Also, a positive correlation between Se and arachidonic acid levels in breast milk was found, linked with upregulated miR-125b. The results of this work demonstrate that miRNA levels are dynamic, probably with the aim of adapting to the specific needs of the infant.

## 1. Introduction

The influence of breastfeeding on maternal health and in short- and long-term health outcomes of infants is beyond scientific doubt. The World Health Organization (WHO) suggests that investing in the promotion of breastfeeding could save thousands of lives and increase the wealth of nations by reducing the incidence of diseases in breastfed children. Due to its benefits, the WHO recommends exclusive breastfeeding from birth to 6 months of age, and its continuation with appropriate complementary foods up to 2 years of age or beyond [1].

Breast milk is a complete and balanced nutritional fluid composed by macro and micronutrients along with different bioactive molecules such as long-chain fatty acids, oligosaccharides, bioactive peptides, immunoglobulins, growth factors or microorganisms that promote infant development and maturation as well as modulate immunity [2,3]. Breast milk is key in the development of the infant’s immune system and has a protective effect in immune-related diseases in the long term [4]. It is also an important source of microRNAs (also called miRNAs), non-coding RNA sequences of about 22 nucleotides that act as post-transcriptional regulators of gene expression [5]. These regulatory molecules are present in all maternal biological fluids but are more abundant in milk [6]. Interestingly, many miRNAs present in milk are conserved between mammals, indicating their fundamental role in milk synthesis, immunity and the development of infants [7]. Breast milk miRNAs mainly originate from mammary epithelial cells and are present in cells, fat and skim milk fractions [8,9]. In this sense, a recent study in bovine milk corroborated that miRNAs profiles of mammary gland and milk exosomes are closely related [10]. Breast milk is composed from hundreds to thousands of different miRNAs but the most abundant ones tend to be the same across different individuals [11]. The most abundant miRNAs families are miR-148a-3p, let-7-5p, miR-30-5p, miR-29-3p and miR-200b/c-3p [12].

Circulating milk miRNAs can be packaged within lipid bi-layered nano-vesicles called exosomes. Those vesicles are secreted out by cells and can contain other cellular biomolecules such as mRNA, DNA or proteins, acting as a mechanism of cellular communication by modulating the function of recipient cells [13]. Exosomes could protect the molecules they carry from degradation [14]. This could result in them not being degraded during digestion, crossing the epithelial barrier and reaching different tissues in vivo where they could have a biological effect [15,16]. This is especially important during the early stages of life due to a still immature intestinal permeability barrier, allowing an increased uptake of macromolecules [17]. However, the pasteurization of human milk, a procedure used in milk banks, reduces the miRNA content of exosomes and their immunomodulatory function due to the degradation of the exosomes [18]. However, it is still a matter of intense debate whether the amount of miRNAs absorbed could exert a true biological effect [19].

The miRNAs enclosed in milk exosomes can exert different functions in the mammary gland itself, regulating, for example, the synthesis and transport of nutrients as well as modulating hormone receptors. Additionally, miRNAs can have modulatory effects in breastfed infants. For example, some immune-related miRNAs are enriched in breast milk exosomes [8] and it has been suggested that some of them can be relevant in the prevention of atopic and autoimmune diseases [20]. Other studies have suggested that exosome-derived miRNAs circulating in maternal milk have an important role in the epigenetic regulation of infants by targeting DNA methyltransferases [21]. Also, they can regulate neurogenesis and gastrointestinal maturation and prevent necrotizing enterocolitis [21,22,23]. On the contrary, the level of some miRNAs in breast milk has been related to early neurological [24] and metabolic [25] disorders in the breastfed infant.

Different endogenous and exogenous factors influence breast milk miRNA profile. The most evident is lactation time, as miRNAs profiles differ between colostrum and the first months of lactation, probably due to an adaptation to the evolving infant’s requirements [26,27]. However, so far, published studies have only evaluated breast milk miRNA content up to 6 six months of lactation. The information on miRNA content of human milk after 6 months of lactation is scarce. Milk miRNA profiles have also been shown to vary between women who deliver prematurely and those who reach full term. This fact could pose an adaptation to the specific developmental needs of premature infants [28,29]. Also, other perinatal factors such as C-section [30], stress [31] or maternal overweight/obesity [32] are reflected in breast milk miRNAs.

Maternal dietary habits have a direct effect on breast milk composition. Milk components as fatty acids [33], minerals [34] or even miRNAs can be influenced by maternal diet [35]. Dietary patterns define not only the nutritional status of the mother but also that of the offspring [36]. The Mediterranean Diet (MD) [37] and the Southern European Atlantic Diet (SEAD) [38] are traditional dietary patterns of the Iberian Peninsula, with characteristic peculiarities that differentiate them. Both have been associated with many health benefits. The consumption of fish, fruits and vegetables and milk and dairy, as well as moderate consumption of meats, are characteristic of these healthy patterns. Both MD and SEAD could have several advantages during lactation as they influence breast milk composition [34,39,40]. A previous study found a correlation between diets rich in animal protein and diets rich in plant protein and breast milk miRNA profiles [35]. For example, the dietary ingestion of polyphenols was reflected in the profile of breast milk miRNAs. In the same way, dietary patterns have an influence on gut microbiota, which may also be reflected in the miRNAs present in breast milk [35]. These modifications can influence infant development.

This work aims to evaluate the effect of lactation time on the levels of miRNAs in breast milk and their correlation with milk composition (fatty acids and minerals) and with dietary patterns of the mother. For this purpose, 11 miRNAs and 59 breast milk samples were selected from mothers who had been lactating from 1 to 59 months. The miRNAs were selected based on those that are highly expressed in breast milk, and for which at the time of study design there was evidence of their epigenetic role and their role in infant development and physiology [12,21,41,42]. This is the first work that evaluates miRNA profiles in women who have been breastfeeding for more than one year and also correlates it with various factors such as breast milk composition or the mother’s diet.

## 2. Materials and Methods

### 2.1. Breast Milk Samples

Samples were obtained from a cohort of mothers recruited for a cross-sectional study evaluating the breast milk composition of lactating mothers living in northwestern Spain. The study protocol was approved by the Galician Clinical Research Ethics Committee (approval code 2016/280), and it is registered at ClinicalTrials.gov (NCT03245697). All procedures adhered to the principles of the Helsinki Declaration of 1975, as revised in 2013. Written informed consent was obtained from all participants. Lactating mothers with gestation <36 weeks, metabolic disorder or chronic diseases were excluded. A total of 20–30 mL of milk were collected in a sterile plastic tube before the first morning breastfeeding session using a breast pump. Samples were aliquoted and stored at −80 °C until analysis. A total of 59 breast milk samples from healthy volunteers with healthy offspring were selected, ranging from 1 to 59 months of lactation.

Each volunteer completed a socio-demographic and health questionnaire including data such as age, weight, height, gestational weigh gain, delivery mode, child birth weight, lactation details (mastitis, delivery/gestational problems and tandem breastfeeding), the number and gender of children, current medication, life habits (smokers or non-smokers and alcohol consumption) and socio-demographic factors (nationality, residency and employment). A food frequency questionnaire with more than 60 items (divided by food groups) and adapted to the regional Southern European Atlantic Diet (SEAD) was used for dietary data collection [43]. Summary data are presented in Table 1.

### 2.2. Exosome and miRNA Isolation

Intact breast milk exosomes were isolated once for each sample using the commercial Total Exosome Isolation Reagent (from other body fluids) (Invitrogen^TM^, Life Technologies, Carlsbad, CA, USA). A total of 1 mL of breast milk was centrifuged for 3 min at 16,000× *g* and the resulting upper fat layer was removed using a spatula. Then, the middle layer was transferred to a new tube and centrifuged for 10 min at 10,000× *g*. After that, 200 μL of clarified milk was pipetted into a new tube without disturbing the pellet, and 200 μL of PBS were added. Finally, 400 μL of Total Exosome Isolation Reagent were added to the tube and the manufacturer’s protocol was followed. The exosomes obtained were resuspended in 50 μL of PBS and preserved at −20 °C until use. The Total Exosome RNA & Protein Isolation Kit (Invitrogen^TM^, Life Technologies) was used to isolate microRNAs from exosomes. The organic extraction and the enriching for small RNAs protocols from the manufacturer’s instructions were followed. RNA samples were stored at −20 °C until use.

### 2.3. Reverse Transcription and qPCR

A total of 11 miRNAs were determined in the present study (Table 2). Two types of microRNA assays were used, TaqMan microRNA assays and TaqMan advanced microRNA assays based on the availability of predesigned assays (Table 2). For the first type of assays, the TaqMan™ MicroRNA Reverse Transcription Kit (Applied Biosystems, ThermoFisher Scientific, Waltham, MA, USA) was used to synthetize cDNA molecules. For the advanced assays, cDNA was synthetized from microRNA isolated from breast milk exosomes using the TaqMan™ Advanced miRNA cDNA Synthesis Kit (Applied Biosystems, ThermoFisher Scientific) following the manufacturer's instructions.

The qPCR assays were performed by triplicate in a QuantStudio 12K Flex real-time PCR system (Applied Biosystems, ThermoFisher Scientific). Reactions were composed of 10 μL of TaqMan^®^ Fast Advanced Master Mix (2X), 1 μL of TaqMan™ Advanced miRNA Assay or TaqManTM miRNA assay, 4 μL of RNase-free water and 5 μL of cDNA template. The qPCR thermal profile was 95 °C for 20 s and 40 cycles at 95 °C for 1 s and 60 °C for 20 s.

### 2.4. Mineral Determination by ICP-MS

The determination of mineral breast milk content (Na, K, Ca, P, Mg, Fe and Se) was carried out by the instrumental analysis unit network of infrastructures supporting the research and technological development of the Universidade de Santiago de Compostela by inductively coupled plasma-mass spectrometry (ICP-MS) (Agilent 7700×, Santa Clara, CA, USA). Briefly, 2 mL of breast milk were digested with 2 mL of H_2_O_2_ at 33% (Panreac, Barcelona, Spain) and 8 mL of HNO_3_ at 69% (Hiperpur, Panreac) using a microwave digestion method (Milestone, Ethos1 Plus, Sorisole, Begamo, Italy) of 190 °C for 15 min at 1000 W. Blanks and a certified reference material (reference material 1549 non-fat milk power, NIST) were included in each digestion batch. Working standard solutions were prepared in NO_3_H-H_2_O. Matrix-matched calibration curves (5 points, R^2^ ≥ 0.9999) were used to calculate concentrations for all elements in milk samples.

### 2.5. Breast Milk Fatty Acids Analysis by GC-FID

The method described in Barreiro et al. [33] was used to determine the fatty acid profile of breast milk samples. A total of 10 µL of breast milk was mixed with 2 mL of H_2_SO_4_ (Merck) at 2.5% in methanol (Merck, Darmstadt, Germany) and saved overnight at 4 °C. After that, samples were incubated at 60 °C for 2 h for fatty acid methylation. Then, fatty acid methyl esters (FAMEs) were extracted with 1 mL of n-hexane (Merck) and determined by gas chromatography using a DB-Was capillary column (60 m, 0.25 µm id, 0.25 µm film thickness; Chrom Tech, Richmond, CA, USA) and a 6850 GC system (Agilent Technologies, Palo Alto, CA, USA), equipped with a flame ionization detector (GC-FID). Data were collected by GC ChemStation software version B.03.02 (Agilent Technologies). Standard mixtures of fatty acid methyl esters “F.A.M.E. mix, C4:0 to C24:0” and “PUFA No. 1, marine source”, a “Linoleic acid methyl ester, cis/trans-isomers” mixture, individual fatty acids (cis-9,trans-11 CLA and trans-10,cis-12 CLA isomers) and the internal standard tricosanoic acid (C23:0) obtained from Sigma Aldrich (Madrid, Spain) were used. The standard was prepared in isooctane (Meck) and calibrators in hexane. The chromatogram was reviewed to check proper automatic peak integration and identification. The percentage of fatty acids by weight was calculated by dividing the peak area for a particular fatty acid by the total sum of the peak areas for all identified fatty acids. All samples were analyzed in duplicate and mean values were used for the study. A total of forty-two fatty acids were identified. The total of saturated fatty acids (SFAs) resulted from the sum of the individual saturated fatty acids: C6:0, C8:0, C10:0, C11:0, C12:0, C13:0, C14:0, C15:0, C16:0, C17:0, C18:0, C20:0, C22:0 and C24:0. For the total monounsaturated fatty acids (MUFAs), the included fatty acids were C14:1n-5, C16:1n-9, C16:1n-7, C16:1n-5, C16:1n-13t, C17:1n-9, C18:1n-9, C18:1n-7, C20:1n-11, C20:1n-9, C22:1n-11 and C22:1n9 and for the total of polyunsaturated fatty acids (PUFAs), the fatty acids included were C18:2n-6, C18:2n-6 9-12t, C18:2n-6 9t-12, CLA18:2n-7 9t-11t, CLA18:2n-6 10t-12, C18:3n-6, C20:2n-6, C20:3n-6, C20:4n-6, C18:3n-3, C18:4n-3, C20:3n-3, C20:4n-3, C20:5n-3, C22:5n-3 and C22:6n-3.

### 2.6. Data Analysis and Statistics

qPCR data were analyzed using the LinRegPCR software (version 2017.0, J.M Ruijter, Amsterdam, The Netherlands) [44,45]. Briefly, non-baseline-corrected data are imported and the software performs a baseline correction on each sample. After that, a window of linearity is determined, and a linear regression analysis is performed to obtain the PCR efficiency per sample from the slope of the regression line. The corrected Cq value obtained was used for the subsequent statistical analysis. The geometrical mean of all Cq values obtained for each miRNA in the study was used to normalize the data of those miRNA.Relative abundance = Cq target miRNA − Cq miRNA geometrical mean

Statistical analyses were performed with GraphPad Prism 10 (GraphPad, Dotmatics, CA, USA). The Kolmogorov–Smirnov test was used to determine the normality of the data. To evaluate the effect of lactation on miRNAs, dietary patterns, fatty acids and minerals levels, the samples were divided into 4 groups, according to lactation time: 1–2 months (*n* = 19), 3–5 months (*n* = 15), 6–11 months (*n* = 9), and 12–59 months (*n* = 16). To perform the comparison between groups, a one-way ANOVA was performed using the two-stage step-up method of Benjamini Krieger and Yekutieli for the false discovery rate (FDR). The library Hmisc in R 4.4.2 was used for the Spearman correlation and adjusted *p*-values by applying Benjamini–Hochberg for the FDR. The correlation between miRNAs and maternal factors, diet, milk fatty acids and minerals was evaluated.

## 3. Results

### 3.1. MiRNA Evolution with Lactation Time

A total of 11 miRNAs were evaluated in 59 breast milk samples, ranging from 1 to 59 months of uninterrupted lactation (Table 3). has-miR-155-5p was not detected in any sample, while hsa-miR-92a was the only miRNA detected in all of the samples included in the study. The hsa-miR-148a-3p presented the lowest Cq mean (24.56 ± 3.46) and hsa-miR-181a-3p the highest Cq mean (32.89 ± 1.68) amongst all samples. The hsa-miR-181a-3p and hsa-miR-30a-5p were only detected in 32 and 20% of all samples, respectively. The percentage of detection varied according to the time of lactation. Between 1 and 3 months of lactation, six miRNAs were detected in all of the samples and hsa-miR-30a-5p was only detected in half of the samples (52.17%). In addition, except for hsa-let-7f-5p, the highest detection rates for all of the assayed miRNAs were always in this group of samples. In group 3–5 months, only hsa-miR-92a was detected in all of the samples while in the groups for 7–12 months and ≥12 months, four miRNAs were detected in all of the samples of the groups. Also, hsa-miR-181a-3p was not detected in the group 7–12 months. Therefore, these results show the effect of lactation time on miRNA profiles.

To determine the relative abundance of each miRNA, the data were normalized using the Cq geometrical mean of all of the miRNAs as an endogenous control. The Kruskal–Wallis test showed significant differences between groups for hsa-miR-148a (Figure 1). The relative abundance of this miRNA was higher in the 1–3 months group and ≥12 months group with respect to the other two groups. Therefore, the abundance of this miRNA tends to decrease with lactation time but after 12 months of lactation it tends to be more similar to that of the first months of lactation. The relative abundance of hsa-miR-92a was also influenced by lactation time. In the group of samples between 1 and 3 months there was less dispersion in comparison with the rest of the groups (Figure 1), suggesting lower interindividual variation in miR-92a in milk during the first months of lactation. Also, there were significant differences between the 1–3 months group and the 6–11 months group, the latter presenting a lower relative abundance of this miRNA. In the case of hsa-miR-125b-5p, there was no significant difference between groups, but it was detected in all samples of the group only in the 1–3 months and ≥12 months groups. For hsa-miRNA-29b-3p, the relative abundance was lower in the group between 6 and 11 months in comparison with the other groups.

The relative abundance of hsa-miR-200c-3p was not significantly different between groups but there were differences in the number of samples in which this miRNA was detected. While in the 1 to 3 month group hsa-miR-200c-3p was detected in approximately 90% of the samples, this percentage decreased to approximately 30% in the 4 to 6 months and 7 to 12 month groups and then increased again to 76% in the group ≥12 months. Therefore, there seems to be a difference in the presence of this miRNA between groups. The hsa-miR-30a-5p was only detected in the group of breast milk samples between 1 and 3 months of lactation. There were substantial abundance differences among the three members of the let-7 family included in this study. While hsa-let-7a-5p and hsa-let-7b-5p were detected in more than 90% of samples, hsa-let-7f-5p was detected only in the 61% of samples. Additionally, the relative abundance of hsa-let-7b-5p is significantly higher during the first 3 months than later in lactation. In the case of has-let-7a-5p, there were significant differences in the groups between 0 and 6 months and the other two groups, showing higher relative abundances earlier in lactation than after 6 months. Therefore, the abundance of these three miRNAs from the let-7 family tends to decrease with lactation time (Figure 1).

Including all of the samples of this study in the analysis, only a significant correlation between miRNA levels and lactation time was observed for hsa-let-7a-5p (*p* = 0.0099, r = 0.4314) and obviously for hsa-miR-30a-5p, and it was mainly detected in the 1–3 months group. A correlation analysis was made without the group of ≥12 months to determine the effect of time. Excluding those samples, a significant correlation between all miRNAs and time was observed, except for hsa-let-7f-5p. The hsa-miR-148a-3p (*p* = 4.24 × 10^−5^, r = 0.5977), hsa-miR-200c-3p (2.46 × 10^−5^, r= 0.6146) and hsa-miR-125b-5p (*p* = 2.83 × 10^−5^, r = 0.6094) presented the most significant correlation with lactation time.

### 3.2. Influence of Maternal Dietary Factors, Fatty Acids and Minerals in miRNA Levels

Table 4 shows the dietary patters of the different groups. No significant differences were observed between groups after ANOVA analysis and post hoc with FDR. Even so, it should be noted that fruit consumption was higher in group 4–5 months in comparison with other groups. In the case of diet, only a higher consumption of fruit was correlated with a higher Cq for hsa-miR-200c-3p (*p* = 0.0214, r = 0.4555) (Figure 2). There was no other correlation between the consumption of certain food groups and miRNA levels.

Fatty acids profiles on the different groups are shown in Table 5. Miristic acid (C14:0), was significantly higher in the >12 months group in comparison with the other groups. Also observed were significant differences in arachidic acid (C20:0), gondoic acid (C20:1 (n-9)), eicosadienoic acid (C20:2 (n-6)) and dihomo-γ-linolenic acid C20:3 (n-6), AA (C20:4 (n-6)) percentages. A higher percentage of some milk fatty acids was correlated with a lower Cq value of certain miRNA (Figure 2). There was also a significant correlation between caprylic acid and hsa-miR-92a (*p* = 0.0485, r = −0.3649). Some miRNAs such as hsa-miR-148a-3p, hsa-miR-29b-3p, hsa-miR-200c-3p and hsa-let-7b-5p showed high correlation with AA abundance in breast milk. Also, hsa-miR-148a-3p, hsa-miR-200c-3p, hsa-miR-125b and has-miR-92a showed a correlation with DHA (C22:6 (n-3)) levels in breast milk. The correlation found between hsa-miR-125b (*p* = 0.0016, r = −0.4935) and AA is particularly relevant.

In the case of minerals, there were differences observed between groups for potassium. The concentration of this miRNA was significantly higher in the 1–3 months group in comparison with the 4–5 months and >12 months groups (Table 6). Potassium levels were negatively correlated with the Cq value of hsa-miR-125b-5p, hsa-let-7b-5p and hsa-miR-92a. It is noteworthy that all of the miRNAs evaluated, with the exception of hsa-let-7a-5p and hsa-miR-30a-5p, showed a high positive correlation with selenium levels (Figure 2).

## 4. Discussion

The results obtained in this study demonstrate that the lactation period influences microRNA levels. Hicks et al. [26] evaluated 147 breast milk samples at 0, 1 and 4 months post-delivery, and observed that most miRNAs (54%; 111/206) decreased in level as lactation progressed, especially those more abundant in breast milk. The hsa-miR-148a-3p, which had the lowest average Cq in the present study, is the most abundant miRNA in any fraction of breast milk, as previously reported [12]. The decrease in the levels of this miRNA as lactation progresses was previously observed in other studies [26,46]. Chiba et al. [46] also observed that hsa-miR-148a-3p levels were higher in transition milk than in mature breast milk from Japanese women. Ahlberg et al. [47] found the same tendency by comparing colostrum with mature milk. It is particularly relevant that the results show that hsa-miR-148a-3p levels in extended lactation milk return to levels similar to those of milk from the first three months of lactation. The presence of this miRNA in milk has been shown to have a positive effect on the newborn [48], along with showing an association with infant growth and adiposity [49]. It plays an important role in epigenetic regulation since there is strong experimental evidence that hsa-miR-148-3p targets the mRNA of the *DNMT1* gene [50]. This gene encodes the enzyme DNA Methyltransferase 1, which is responsible for maintaining DNA methylation patterns by transferring methyl groups to the cytosine nucleotide. A study carried out in mice that received porcine milk exosomes demonstrated that they increased the expression of this miRNA in colorectal cells while decreasing the expression of DNMT1 [51]. hsa-miRNA-148a-3p seems to play an important role in inflammation at the intestinal level since it can suppress signals associated with inflammation, and in patients with inflammatory bowel disease this miRNA was downregulated [52]. hsa-miRNA-148a-3p *DNMT1* mRNA targeting increases the expression of the master transcription factor of regulatory T cells (Tregs), FoxP3, enhancing their expansion for the control of intestinal immune tolerance and eventually systemic immune tolerance development.

The presence of this miRNA in breast milk could indicate the important role of breast milk as a messenger between mother and infant as it could play a key role in the regulation of gene expression. Notably, the deficiency of exosomal miRNA in infant formula could result in epigenetic disturbance with long-term consequences for bottle-fed infants [41].

The miR-92a was one of the miRNAs most influenced by lactation time in the breast milk lipid fraction from U.S. mothers [26]. It is a ubiquitous regulator of B-cell, T-cell and monocyte development with an important immunomodulatory role [5]. Its presence in breast milk can be important in the development of the infant’s immune system, especially in the first months of life, which is why it decreases as lactation progresses and the infant’s immune system matures. Interestingly, the overexpression of miRNA-92a-3p in skeletal muscle cells has been suggested as a protecting mechanism in gestational diabetes, as it increases insulin-stimulated glucose uptake [53]. It makes sense then that its levels decrease gradually in body fluids after delivery, as the risk of gestational diabetes also goes away.

In breast milk lipid fraction samples, hsa-miR-125b-5p was slightly affected by lactation time [26] and Chiba et al. (2022) [46] also observed a difference in the levels between transition and mature milk. This miRNA has been demonstrated to have a beneficial effect in the development of infant regurgitation by contributing to the survival and proliferation of gastrointestinal cells and maintenance of the gastrointestinal barrier [54]. Also, miR-125b attenuates p53 activity by directly suppressing *TP53* mRNA (transcriptional mechanism), while miR-148a does the same by promoting proteasomal p53 degradation (posttranscriptional mechanism) [23,41,55]. The p53 is a transcription factor acting as a key tumor suppressor and a central regulator of the stress response, also known as the “guardian of the genome”. This negative regulation of p53 by both miRNAs is highly relevant to the newborn’s neurological development and growth, as it represses apoptosis in human cells and supports normal tissue growth. Curiously, miR-148a and miR-125b have shown similar patterns across breast milk samples, as depicted in Figure 1, supporting the theory of them co-working for p53 regulation. Hicks et al. found that hsa-miRNA-29b-3p was in the top 25 of those affected by the lactation period [26]. This miRNA is important in fetal neurogenesis and osteoblast differentiation [56,57]. Higher levels of hsa-miRNA-29b-3p have been observed in colostrum for preterm infants [28], while those babies usually show lower circulating miR-29b in their plasma than term counterparts. In this sense, pulmonary complications of preterm infants are frequent, and decreased circulating miR-29b is inversely correlated with pulmonary disease severity. One could think that the higher levels of miR-29b in colostrum are to compensate for the child’s deficiencies and protect it from preterm complications, supporting the idea of encouraging breastfeeding of the preterm baby This previous fact is in agreement with previous findings reported by Durrani-Kolarik et al., who demonstrated a restorative effect of miR-29b at the pulmonary level in a mouse model mimicking the pathophysiology observed in infants with severe bronchopulmonary dysplasia [58].

Hicks et al. [26] found that the hsa-miR-30 family is highly affected by lactation time, downregulating its levels in the breast milk lipid fraction as lactation evolves. That same trend is observed in this study as this miRNA was only detected in the 1–3 month samples. The miR-30a-5p has an important role in signaling pathways to alleviate inflammation in intestinal epithelial cells, which may be important during the first stages of life to protect against inflammatory attack [59]. In a study carried out with goat milk exosomes it was observed that miR-30a-5p could reduce LPS-induced intestinal inflammation in a mouse model [60]. Therefore, this miRNA could have a protective effect in the first steps of intestinal colonization by bacteria, avoiding inflammation caused by bacterial LPS. Also, miR-30a in human milk has been related to increased infant weight gain at 12 months of age, as it targets pathways involved in glucose homeostasis and metabolism [25]. Similarly, hsa-miR-181a-3p was only detected in 32% of the samples analyzed in this study, being present in 56% of the samples between 1 and 3 months and with none or very low amplification rates in the rest of the groups. It was observed that this miRNA was one of those most affected by lactation time, and it is detected in higher abundance in colostrum than in mature milk [26,61]. It has also been shown to be upregulated in the mature breast milk of mothers with very/moderately preterm infants [28]. Interestingly, miR-181a expression is negatively correlated with leptin milk levels [32]. As leptin regulates appetite, a negative association between breast milk leptin and infant weight gain has been observed [62]. Therefore, its decreased levels in human milk would result in an increased appetite, food intake, and weight gain of the preterm infant. This is also one of those miRNAs called immuno-related miRNAs as it regulates the process of the maturation and differentiation of B and T lymphocytes [5]. In the case of the breast milk lipid fraction, Hicks et al. [26] found that the level of miR-200c was significantly affected by lactation time. Similarly, a study carried out in ovine mammary gland found a higher expression of this miRNA at late pregnancy and early lactation and then a decrease in mid and late lactation [63]. In the infant, it could be related to gut maturation by modulating epithelial function and exosomal endocytosis, and therefore, the uptake of other miRNAs [64]. Interestingly, overexpressed miR-200c has been found in serum from children with Kawasaki disease, suggesting it also has a role in the inflammatory response [65].

The let-7 family was influenced by lactation period. This was especially relevant for hsa-let-7a-5p, which showed a high correlation with lactation time. This family is implicated in the expression of growth hormone, estrogen and in the prolactin signaling pathway [64]. Those hormones play a role in the production of milk and would indicate the importance of these miRNAs in the regulation of lactation. Also, the evaluated miRNAs have immunomodulatory functions that can regulate an infant’s development [7], and implications in oral tolerance development [12]. In particular, hsa-let-7a-5p is highly expressed in very/moderately preterm breast milk, a fact that could indicate that this miRNA has an important role in the development of the nervous system as well [28].

Previous works have demonstrated that the type of diet could influence the miRNA profile of breast milk [26,35,47]. However, in this study, only a slight correlation could be found with fruit consumption. Fruit is a source of polyphenols and previous studies have observed a correlation between breast milk microRNAs and these bioactive molecules [35]. In future studies, it will be interesting to determine whether it is the polyphenols themselves that directly influence miRNA expression in cells or whether it is the diet associated with polyphenol consumption that actually influences miRNA levels. Shah et al. found an association between diet quality/healthy eating index and an increased abundance of miR-148a, miR-30b, miR-let-7a and miR-let-7d, with dairy being an important component of this index [49]. Likewise, in a previous study, differences in miRNAs in breast milk related to cell proliferation have been observed between diets rich in vegetable protein and diets rich in animal protein, being overexpressed in the latter [35]. Although not significant, this study showed some correlation between miR-29b and the Southern European Atlantic diet (*p* = 0.0585, r = −0.3705), a dietary pattern characterized among other things by a high consumption of dairy products. In this sense, Baier et al. (2014) observed that milk consumption increased postprandial levels of miR-200c and miR-29b, which could indicate an absorption of those miRNAs present in milk. However, there is still controversy regarding the absorption of significant amounts of miRNAs through the diet [66]. The miRNAs evaluated for Baier et al. (2014) are conserved in *Bos taurus* and humans. One of the keys is to identify whether the increase in circulating levels of these miRNAs may be due to their absorption or whether they increase their expression in humans due to a direct/indirect effect of dietary components [66]. Conversely, Witwer [67] did not observe variations in these miRNAs in relation to milk consumption. López de las Hazas et al. (2022) loaded exosomes with five frequently reported breast milk miRNAs (including miR-148a-3p and miR-30a-5p) in extracellular vesicles (EVs) and observed, in a mice model, that after digestion, those miRNAs reached host organs, including the brain [15]. It should be noted that cow milk consumption-associated bovine milk miR uptake in adult human volunteers is not comparable to the postnatal period characterized by the higher intestinal permeability of newborn infants. In this sense, Weil et al. corroborated, in a porcine model, the vertical transmission of milk miRNAs reaching the systemic circulation during neonatal time periods [68].

Many correlations were found between miRNA levels and fatty acids. In this sense, it is a known fact that some miRNAs are involved in the regulation of mammary gland fatty acids synthesis pathways [21]. In the synthesis of long chain fatty acids such as arachidonic acid (AA), eicosapentaenoic acid (EPA) or docosahexaenoic acid (DHA), enzymes related to β-oxidation are involved, such as desaturases or elongases [69]. Likewise, a negative correlation of miR-148a and saturated fatty acids was found in bovine milk, along with a positive correlation to the monounsaturated fraction and unsaturated/saturated ratio [70]. These results may indicate a relevant role of these miRNAs in the synthesis of fatty acids in the mammary gland. The correlation found between hsa-miR-125b (*p* = 0.0016, r = −0.4935) and AA is particularly relevant. This miRNA can target *ELOVL* mRNAs, which encode for elongases that are implicated in the synthesis of AA from its precursor, linoleic acid. Also, 5-Lipoxygenase, an enzyme responsible for the synthesis of leukotrienes from ARA, is a direct target of hsa-miR-125b [69]. This miRNA causes a repression of the expression of the enzyme by binding to the mRNA [71]. In the case of miR-200c-3p, a study carried out in ovine mammary epithelial cells found that this miRNA is involved in fatty acid synthesis by targeting the *PANK3* gene [63]. A high expression of miR-200c-3p results in the inhibition of the *PANK3* gene and higher synthesis of triglycerides. But also, some fatty acids can upregulate the expression of some miRNAs. For example, it has been observed that EPA upregulates the expression of miRNA miR-30b-5p, which differs only in two nucleotides in the seed sequence compared to miR-30a-5p. This miRNA is involved in the development of brown adipose tissue and thermogenesis, which is of importance for the newborn and especially for the preterm infant [72,73,74]. Likewise, the type of fat ingested in the diet can also have a direct effect on the expression of miRNAs. In a study in pregnant rats, it was observed that the type of oil used in the diet modified the expression of certain microRNAs in both maternal and offspring tissues. This would indicate the epigenetic effect of the maternal diet on the offspring [75].

One of the major novelties of the present study is the evaluation of the association between the levels of miRNAs and minerals in breast milk. Thus, higher selenium concentrations were correlated with lower Cq, which would indicate a higher level of miRNAs in breast milk. Selenium is an essential mineral that plays various functions in the body including the modulation of the immune system, antioxidant activity, and free radical scavenging [76]. In addition, selenium is part of selenoproteins, which play a fundamental role in the regulation of gene expression. Various studies have shown that selenium levels have an important influence on the expression of miRNAs. Feng et al. [77] found that deficiency/supplementation with selenium altered the expression profile of 119 miRNAs in a rat model. Focusing on the eleven miRNA evaluated in this study, Liu et al. observed that low levels of selenium were related to a down-regulation of miR-29a-3p in pig brain [78]. On the other hand, Chen et al. (2016) observed that moderate levels of selenium are related to an up-regulation of miR-125b [79]. As discussed above, miR-125b was positively correlated with AA levels in the samples analyzed in this study. It is interesting to note that certain pathways such as arachidonic acid metabolism are selenium sensitive [80]. In future studies, it would be interesting to evaluate the effect of selenium levels on the miRNome of breast milk. This would give an idea of how the levels of this mineral influence the information that is transmitted to the infant through exosomes. It is also important to note that factors such as hydration can have an influence on the electrolytes in milk, so these results should be taken with caution.

This work has certain limitations. The number of samples included is low and it was not a longitudinal study of the volunteers but rather a cross-sectional observational study. This limits the comparison between different lactation periods. It is important to mention that the extended lactation group only has 16 volunteers with a very wide range of lactation time, and therefore, the conclusions drawn are limited. The limited number of samples in each group reduces the statistical power of this study, which may result in some differences or correlations between microRNAs and diet and maternal factors not being detected. A lower number of samples per group could increase the variability and influence the ANOVA and correlation analysis. Another limitation of the study is the high Ct obtained for several of the miRNAs, which has resulted in the fact that some of them could not be detected in all of the samples analyzed. This fact limits the statistical results obtained between lactation groups and the correlations obtained between miRNA levels and maternal diet and milk composition. In addition, in this study, a targeted analysis of miRNAs was performed, so many potential correlations between breast milk miRNAs and maternal factors and diet could not be detected. One of the most common methods used to normalize gene expression data obtained by qPCR is the use of housekeeping genes. In the case of miRNAs it is highly complex to find housekeeping miRNAs. One of the options that can be used to normalize these data is to spike miRNAs into the samples. In this work, because we did not used spiked miRNAs, we opted to do a normalization of the data based on the geometric mean of all miRNAs. This is a technique more commonly employed in RNA-seq where there are thousands of miRNAs expressed. Although the method employed is not common in qPCR and has limitations such as not taking into account the individual variability of each sample, it may represent an alternative to comparing the expression of an miRNA in a sample with respect to a geometrical mean of all miRNAs. Therefore, although our results show some variation in miRNAs with respect to lactation time or correlation with some maternal factors, a definitive significance cannot be obtained from the present study. Regarding the technical aspect, only one commercial kit for the extraction of exosomes has been tested and purity analyses have not been performed after their isolation. Therefore, these exosomes may be contaminated with proteins or other molecules. In a recent study, the minimal information for the study of extracellular vesicles was defined [81]. On the other hand, the absence of some housekeeping miRNAs or the addition of exogenous miRNAs during the isolation process for normalizing the results limits the scope of the results.

## 5. Conclusions

The importance of breast milk miRNAs in infant development is something that is not fully understood. Their high bioactivity, the complex population of these molecules and their packaging in exosomes seem to indicate that they are designed to act as mother/child messengers. But factors such as their absorption at the intestinal level or whether they are absorbed in sufficient quantities are still a matter of debate. The results of this study show that miRNA profiles evolve over time during lactation, which would indicate an adaptation to the development of the child, as does the fatty acid profile of milk and hence the possible correlation between both elements.. It will be interesting to corroborate these results and their biological significance in future studies. With the exception of a slight correlation with fruit consumption, this study was unable to determine whether diet could have an effect on microRNA levels in milk.

## Figures and Tables

**Figure 1 foods-14-01003-f001:**
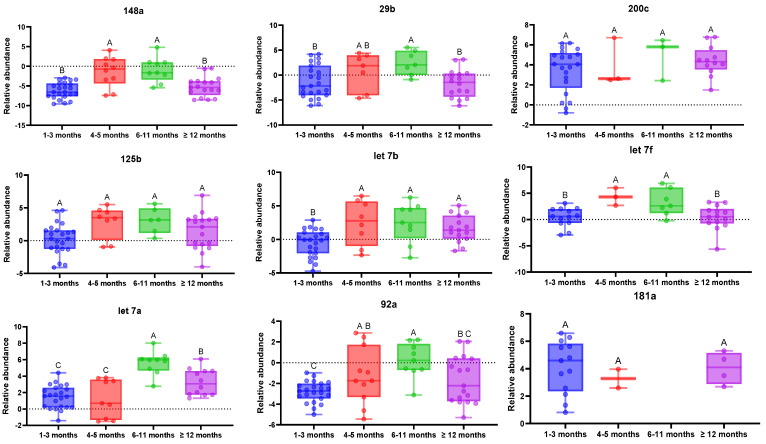
Box plots showing the evolution of breast milk miRNAs in different periods of lactation, from 1 to 59 months. Different letters between groups indicate significant differences (*p* < 0.05). The Cq value is inversely proportional to the initial amount of miRNAs in the sample. If the Cq value is lower than the geometric mean it indicates higher initial microRNA concentration. In these cases, the result of the applied formula is negative. Therefore, a higher negative value indicates a higher initial concentration of miRNA in the sample.

**Figure 2 foods-14-01003-f002:**
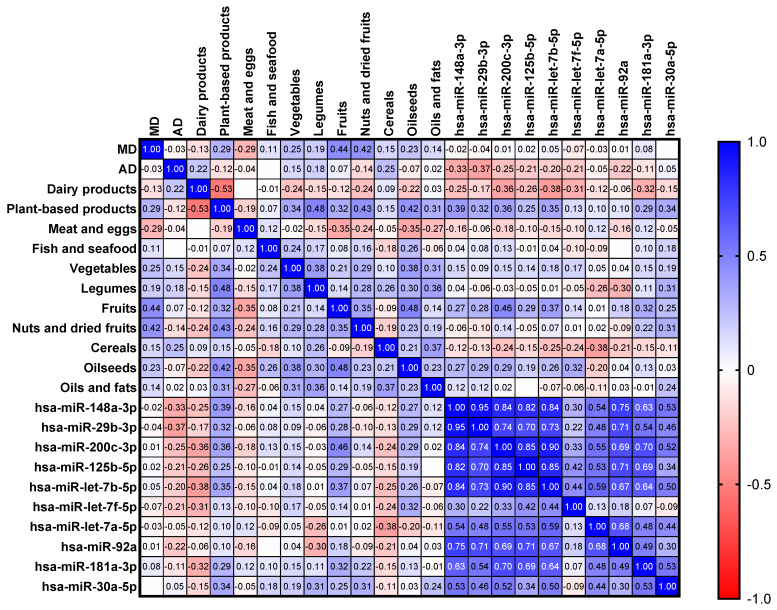
Spearman’s rank correlation matrix between miRNAs, maternal dietary factors and fatty acids and minerals in breast milk.

**Table 1 foods-14-01003-t001:** Anthropometric, pregnancy, delivery and diet data characteristics of volunteers.

	Mean	Standard Deviation
Age (years)	35.33	4.27
Weight (kg)	64.69	10.74
Height (m)	1.65	0.06
BMI (kg/m^2^)	23.80	3.66
Gestational age at birth (weeks)	39.72	1.29
Weight gain (kg)	13.32	3.71
Infant weight (kg)	3.35	0.45
Lactation time (months)	10.76	13.36
	%	
Infant gender: male (%)	49.12	
Infant gender: female (%)	50.88	
Caesarean delivery (%)	13.46	
Parity number: 1st child (%)	61.40	
Parity number: 2nd child (%)	36.84	
Parity number: 3rd child (%)	1.75	
Tandem breastfeeding (%)	8.47	
	Mean	Standard deviation
Adherence to MD (0–14)	8.30	1.87
Adherence to SEAD (0–9)	3.89	1.41
Dairy products (servings/day)	3.23	1.75
Plant-based non-dairy products (servings/day)	0.27	0.75
Meat and eggs (servings/day)	1.96	0.98
Fish and shellfish (servings/day)	0.85	0.44
Vegetables (servings/day)	4.52	3.18
Legumes (servings/day)	0.2	0.22
Fruits (servings/day)	4.36	3.09
Nuts and dried fruit (servings/day)	0.64	1.00
Cereals (servings/day)	2.81	1.39
Oils and fats (servings/day)	2.35	1.31

**Table 2 foods-14-01003-t002:** Assay characteristics for eleven miRNAs analyzed in breast milk samples.

microRNA	Assay Design	Assay Code *
hsa-miR-148a-3p	Advanced	477814_mir
hsa-miR-29b-3p	Advanced	478369_mir
hsa-miR-125b-5p	Advanced	477885_mir
hsa-miR-200c-3p	Advanced	478351_mir
hsa-let-7f-5p ¥	Advanced	478578_mir
hsa-let-7b-5p ¥	Advanced	478576_mir
hsa-let-7a-5p ¥	Normal	000377
hsa-miR-92a	Normal	000431
hsa-miR-181a-3p	Advanced	479405_mir
hsa-miR-30a-5p	Advanced	479448_mir
hsa-miR-155-5p	Advanced	483064_mir

* Assay code from ThermoFisher Scientific. ¥ The let family was first discovered in the nematode *Caenorhabditis elegans* and named after the gene that synthesized them. Because of this, they do not carry the abbreviation miR in their name.

**Table 3 foods-14-01003-t003:** Descriptive statistics of miRNA detection by qPCR in breast milk from Spanish mothers (*N* = 59) and amplification rates according to lactation time.

					Amplification (% of Samples)				
microRNA	Cq Mean	Cq SD	Cq Min	Cq Max	All Samples	1–2 Months	3–5 Months	6–11 Months	≥12 Months
hsa-miR-148a-3p	24.56	3.46	19.28	33.66	98.31	100.00	90.91	100.00	100.00
hsa-miR-29b-3p	28.18	3.36	22.72	34.4	88.14	100.00	63.64	77.78	94.12
hsa-miR-125b-5p	30.18	2.64	24.79	35.78	89.83	100.00	72.73	66.67	100.00
hsa-miR-200c-3p	32.5	2.63	21.92	35.65	67.80	91.30	27.27	33.33	76.47
hsa-let-7f-5p	30.1	2.54	23.23	35.74	61.02	56.52	27.27	77.78	82.35
hsa-let-7b-5p	29.59	2.84	22.74	35.34	94.92	100.00	72.73	100.00	100.00
hsa-let-7a-5p	31.32	2.35	26.1	36.89	91.53	100.00	81.82	100.00	76.47
hsa-miR-92a	27.11	2.14	23.42	31.73	100.00	100.00	100.00	100.00	100.00
hsa-miR-181a-3p	32.89	1.68	29.68	35.46	32.20	56.52	18.18	0.00	23.53
hsa-miR-30a-5p	27.3	2.03	24.53	30.69	20.34	52.17	0.00	0.00	0.00
hsa-miR-155-5p	-	-	-	-	0.00	0.00	0.00	0.00	0.00

**Table 4 foods-14-01003-t004:** Dietary patterns and adherence to Mediterranean and Southern European Atlantic diet in the different groups included in the present work.

	Group			
	1–3 Months	4–5 Months	6–11 Months	>12 Months
Dairy products (servings/day)	3.48 ± 1.62	2.46 ± 1.59	3.35 ± 0.69	3.3 ± 2.19
Plant-based non-dairy products (servings/day)	0.25 ± 0.92	0.4 ± 0.52	0.04 ± 0.08	0.27 ± 0.7
Meat and eggs (servings/day)	2.18 ± 0.89	1.49 ± 0.63	1.4 ± 1	2.1 ± 1.16
Fish and shellfish (servings/day)	0.85 ± 0.36	0.92 ± 0.7	0.6 ± 0.37	0.88 ± 0.41
Vegetables (servings/day)	3.65 ± 1.96	6.41 ± 4.68	3.85 ± 2.04	4.81 ± 3.55
Legumes (servings/day)	0.15 ± 0.1	0.3 ± 0.29	0.1 ± 0.04	0.25 ± 0.3
Fruits (servings/day)	3.81 ± 2.81	7.24 ± 3.36	3.5 ± 2.64	3.69 ± 2.74
Nuts and dried fruit (servings/day)	0.32 ± 0.49	0.64 ± 0.81	1.18 ± 1.96	0.91 ± 1.25
Cereals (servings/day)	2.8 ± 1.21	2.65 ± 1.32	1.89 ± 1.32	3.16 ± 1.66
Oils and fats (servings/day)	2.09 ± 1.22	2.61 ± 1.28	2.24 ± 1.31	2.59 ± 1.5
Adherence to MD (normalized to 0-1)	0.57 ± 0.14	0.65 ± 0.09	0.59 ± 0.16	0.6 ± 0.14
Adherence to SEAD (normalized to 0-1)	0.42 ± 0.16	0.36 ± 0.09	0.46 ± 0.12	0.48 ± 0.19

**Table 5 foods-14-01003-t005:** Fatty acids profiles (% wt/wt of total fatty acids) in breast milk. Different letters in the same row indicate significant differences.

	Group			
	1–3 Months	4–5 Months	6–11 Months	>12 Months
C6:0	0.471 ± 0.315	0.571 ± 0.365	0.548 ± 0.473	0.239 ± 0.113
C8:0	0.308 ± 0.083	0.256 ± 0.046	0.224 ± 0.037	0.284 ± 0.055
C10:0	1.778 ± 0.454	1.595 ± 0.337	1.412 ± 0.271	1.696 ± 0.274
C11:0	0.045 ± 0.018	0.053 ± 0.022	0.046 ± 0.033	0.039 ± 0.016
C12:0	9.711 ± 4.215	8.657 ± 1.83	8.546 ± 1.143	11.043 ± 1.829
C13:0	0.038 ± 0.011	0.032 ± 0.012	0.032 ± 0.012	0.046 ± 0.013
C14:0	6.373 ± 2.436 ^B^	6.484 ± 1.99 ^B^	6.999 ± 1.27 ^B^	10.281 ± 1.986 ^A^
C14:1 (n-5)	0.222 ± 0.11	0.189 ± 0.064	0.168 ± 0.07	0.207 ± 0.119
C15:0	0.255 ± 0.078	0.213 ± 0.085	0.208 ± 0.093	0.216 ± 0.076
C16:0	19.578 ± 2.217	18.07 ± 3.102	17.788 ± 4.34	17.518 ± 2.227
C16:1 (n-9)	0.575 ± 0.133	0.513 ± 0.081	0.476 ± 0.108	0.575 ± 0.156
C16:1 (n-7)	2.406 ± 0.891	2.196 ± 0.81	1.969 ± 0.682	2.238 ± 0.839
C16:1 (n-5)	0.071 ± 0.029	0.057 ± 0.012	0.054 ± 0.018	0.061 ± 0.027
C16:1 (n-13)t	0.08 ± 0.037	0.072 ± 0.049	0.069 ± 0.032	0.054 ± 0.019
C17:0	0.304 ± 0.055	0.274 ± 0.084	0.267 ± 0.09	0.236 ± 0.049
C17:1 (n-9)	0.224 ± 0.078	0.185 ± 0.056	0.167 ± 0.041	0.201 ± 0.063
C18:0	7.034 ± 1.881	6.624 ± 1.305	6.229 ± 1.75	4.918 ± 0.925
C18:1 (n-9)	27.93 ± 6.531	31.562 ± 6.291	35.889 ± 8.35	27.602 ± 8.369
C18:1 (n-7)	0.731 ± 0.137	0.693 ± 0.124	0.619 ± 0.11	0.695 ± 0.135
C18:2 (n-6)	15.95 ± 4.379	15.919 ± 4.922	13.376 ± 3.718	16.04 ± 4.337
C18:2 (n-6)9,12t	0.159 ± 0.045	0.157 ± 0.045	0.13 ± 0.039	0.163 ± 0.06
C18:2 (n-6)9t,12	0.129 ± 0.036	0.128 ± 0.031	0.11 ± 0.031	0.138 ± 0.049
C18:3 (n-6)	0.127 ± 0.085	0.14 ± 0.053	0.075 ± 0.046	0.086 ± 0.051
C18:3 (n-3)	0.586 ± 0.256	0.676 ± 0.269	0.594 ± 0.175	1.008 ± 0.905
C18:2 (n-7)9,11t	0.609 ± 0.175	0.487 ± 0.14	0.477 ± 0.164	0.539 ± 0.212
C18:4 (n-3)	0.122 ± 0.044	0.13 ± 0.046	0.117 ± 0.064	0.108 ± 0.039
C18:2 (n-6)10t,12	0.341 ± 0.129	0.34 ± 0.105	0.318 ± 0.153	0.315 ± 0.113
C20:0	0.164 ± 0.04 ^A^	0.174 ± 0.041 ^A^	0.146 ± 0.034 ^A,B^	0.122 ± 0.021 ^B^
C20:1 (n-11)	0.068 ± 0.019	0.081 ± 0.04	0.075 ± 0.048	0.077 ± 0.044
C20:1 (n-9)	0.504 ± 0.113 ^A^	0.413 ± 0.091 ^A,B^	0.361 ± 0.059 ^B^	0.41 ± 0.122 ^A,B^
C20:2 (n-6)	0.366 ± 0.084 ^A^	0.275 ± 0.078 ^B^	0.246 ± 0.032 ^B^	0.328 ± 0.091 ^A,B^
C20:3 (n-6)	0.612 ± 0.174 ^A^	0.417 ± 0.084 ^B^	0.32 ± 0.07 ^B^	0.447 ± 0.165 ^B^
C20:4 (n-6)	0.663 ± 0.143 ^A^	0.509 ± 0.103 ^A,B^	0.409 ± 0.089 ^B^	0.639 ± 0.214 ^A^
C20:3 (n-3)	0.066 ± 0.038	0.08 ± 0.055	0.073 ± 0.07	0.055 ± 0.022
C20:4 (n-3)	0.101 ± 0.033	0.1 ± 0.05	0.109 ± 0.095	0.059 ± 0.028
C20:5 (n-3)	0.134 ± 0.095	0.155 ± 0.089	0.133 ± 0.107	0.097 ± 0.063
C22:0	0.071 ± 0.023	0.076 ± 0.03	0.063 ± 0.016	0.059 ± 0.014
C22:1 (n-11)	0.084 ± 0.047	0.093 ± 0.086	0.054 ± 0.025	0.035 ± 0.016
C22:1 (n-9)	0.104 ± 0.026	0.095 ± 0.025	0.088 ± 0.03	0.079 ± 0.02
C22:5 (n-3)	0.137 ± 0.053	0.127 ± 0.033	0.112 ± 0.024	0.148 ± 0.067
C24:0	0.052 ± 0.048	0.075 ± 0.071	0.046 ± 0.031	0.028 ± 0.013
C22:6 (n-3)	0.497 ± 0.354	0.351 ± 0.19	0.328 ± 0.26	0.494 ± 0.298
Total SFA	45.931 ± 6.985	43.078 ± 6.119	42.431 ± 7.569	46.508 ± 3.958
Total MUFA	33.515 ± 6.991	36.927 ± 5.996	40.671 ± 8.099	32.801 ± 7.672
Total PUFA	20.483 ± 4.675	19.92 ± 5.065	16.834 ± 3.788	20.635 ± 5.685
Total PUFA n-3	1.545 ± 0.612	1.562 ± 0.381	1.383 ± 0.565	1.94 ± 1.127
Total PUFA n-6	17.988 ± 4.619	17.532 ± 5.067	14.656 ± 3.67	17.865 ± 4.843
CLAs	0.95 ± 0.251	0.826 ± 0.24	0.796 ± 0.303	0.854 ± 0.315

**Table 6 foods-14-01003-t006:** Mineral levels in breast milk in the four groups.

	Group			
	1–3 Months	4–5 Months	6–11 Months	>12 Months
Na (mg/L)	144.4 ± 69.57	101.15 ± 49.65	109.16 ± 46.91	232.46 ± 188.79
Mg (mg/L)	31.64 ± 6.31	32.74 ± 5.73	34.34 ± 2.4	35.63 ± 8.89
P (mg/L)	133.02 ± 25.91	119.75 ± 10.27	119.32 ± 26.5	132.05 ± 39.49
K (mg/L)	502.54 ± 65.4 ^A^	415.57 ± 32.32 ^B^	451.94 ± 68.39 ^A,B^	420.95 ± 53.92 ^A,B^
Ca (mg/L)	301.61 ± 71.15	267.48 ± 36.73	255.53 ± 17.64	246.85 ± 49.1
Fe (mg/L)	226.72 ± 93.41	196.48 ± 90.22	170.93 ± 96.83	219.98 ± 94.48
Se (µg/L)	12.47 ± 4.22	7.62 ± 1.35	10.36 ± 7.44	12.12 ± 5.12

Different letters in the same row indicate significant differences.

## Data Availability

The original contributions presented in this study are included in the article. Further inquiries can be directed to the corresponding authors.

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
