# Peer review of "Influence of Maternal Diet and Lactation Time on the Exosomal miRNA Cargo in Breast Milk"

_foods, 2025, doi:10.3390/foods14061003_

Round 1
Reviewer 1 Report
Comments and Suggestions for Authors
Authors describe the presence of exosomal miRNAs in breast milk of 59 Spanish mothers during different stages of laction and correlate them with different parameters including diet, minerals, fatty acids and others. The topic is relevant but there are some problems with the analysis.
The topic of how miRNAs changes along lactation is of extraordinary importance. Not only for understanding basic biology of human growth but also for developing future infant formula. The major problem is the design of the study. An appropriate design would have been to follow the same mother along the different periods. This reviewer understand that that is not easy. Al alternative could have been to follow non-consecutive samples of a larger cohort of mothers. The problem of not detecting certain miRNAs in certain samples is not clear whether it is because the are absent or because of technical problem during isolation. Indeed, author have more than 10 ml of milk and only analyze 200 uL of samples.. that is a very limited amount of sample.
Moreover, there is a relevant limitation in the analytical method of miRNA analysis. Authors do not use any spike RNAs or housekeeping (biological fluids do not usually have housekeeping genes) RNAs to normalize data. Instead, they use the geometrical mean of all Cq values obtained for each miRNA in the study was used to normalize the data of those miRNA. Although this might be a valid formula, the amount of samples analyzed is very low. This and other methods based on these type of normalization used hundreds or thousands of samples or miRNAs targets.
Exosome isolation method does not give purified exosomes, but contaminated with many other proteins and molecules. Unless some additional purification is performed and verify by other methos such as electron microscopy, western blot or other, authors should empathize that exosomes are not pure in the limitation section. This has been already acknowledged.
Figure 1 show relative abundance of miRNAs during lactation period. For example, miR-148a, relative abundance is lower <3 months and >12 months. They are in the negative scale. Shouldn’t they be in the positive scale and the other in the negative, so it indicates that their levels are higher (<3 months and >12 months). Is figure 1 (and all the others) correct? If so, please indicate in figure legend how to interpret those data. As it is now, it is misleading.
Authors indicate that the aim of this work is to evaluate the effect of lactation time on the levels of miRNAs in breast milk and their correlation with milk composition (fatty acids, minerals, microbiota) and with dietary patterns of the mother. But no data on microbiota is presented. Please include data of microbiota.
Authors use a previous published cohort of Spain mothers: Sanjulián L, Lamas A, Barreiro R, Cepeda A, Fente CA, Regal P. Bacterial Diversity of Breast Milk in Healthy Spanish Women: Evolution from Birth to Five Years Postpartum. Nutrients. 2021 Jul 14;13(7):2414. doi: 10.3390/nu13072414. PMID: 34371924; PMCID: PMC8308733. However, in this manuscript they use only 59 samples from a total of 99. Please indicate why not the 99 samples were used. They acknowledge that there is a low number of samples, but why not using the whole cohort of 99 samples (although they are still a low number of samples)
Authors do not show any results of the diet, minerals, fatty acids and dietary patterns. These data must be included both in material and methods section and in the results sections. These data are relevant and must be included in this manuscript.
Include the rationale to select those 11 miRNAs and not others.
Authors indicate that the miRNA levels in breast milk after 12 months appear similar to those observed during the first months of lactation. This reviewer is not sure whether this affirmation is correct. With all the technical problems mentioned above, authors should have caution with the conclusions of this work.
L66 modulating the reception of hormones? Please explain or rephrase
Author Response
Comment 1: The topic of how miRNAs changes along lactation is of extraordinary importance. Not only for understanding basic biology of human growth but also for developing future infant formula. The major problem is the design of the study. An appropriate design would have been to follow the same mother along the different periods. This reviewer understand that that is not easy. Al alternative could have been to follow non-consecutive samples of a larger cohort of mothers. The problem of not detecting certain miRNAs in certain samples is not clear whether it is because the are absent or because of technical problem during isolation. Indeed, author have more than 10 ml of milk and only analyze 200 uL of samples. that is a very limited amount of sample.
Response 1: We really appreciate the constructive comments of Reviewer 1. We totally agree, the best way to really evaluate the evolution during lactation would be to make a longitudinal study of the same mothers. This would allow to eliminate the variability of each lactating woman in the analysis. It is also true that in the case of mothers with more than one year of lactation it would be difficult to perform a longitudinal analysis due to the complexity of finding this type of volunteers. In future studies we will try to increase the cohort of mothers to reduce the limitations of this study. In the limitation part we have added the next information:
“The number of samples included is low and it was not a longitudinal study of the volunteers but rather a cross-sectional observational study. This limits the comparison between different lactation periods”
In the case of 200uL, we used the indications of the commercial protocol, which indicated the use of 200uL for milk samples. In future studies we will test whether increasing the amount of sample with this kit will improve the results. This could prevent the non-identification of miRNAs when they are present in low amounts in exosomes.
Comment 2: Moreover, there is a relevant limitation in the analytical method of miRNA analysis. Authors do not use any spike RNAs or housekeeping (biological fluids do not usually have housekeeping genes) RNAs to normalize data. Instead, they use the geometrical mean of all Cq values obtained for each miRNA in the study was used to normalize the data of those miRNA. Although this might be a valid formula, the amount of samples analyzed is very low. This and other methods based on these type of normalization used hundreds or thousands of samples or miRNAs targets.
Response 2: At the beginning of the study we considered the use of some of the microRNAs such as 92a or let7 as controls since some of the studies reviewed indicated that they were stable. Once the results were analyzed, variability was observed between microRNAs. As indicated by the reviewer, the geometric mean of all microRNAs is one of the approaches used in RNA-seq as a normalizer. Due to the amount of data obtained with RNA-seq this normalization allows the use of thousands of microRNAs. The approach we have employed is not the usual one but using the geometric mean of the microRNA to normalize allows to evaluate how the concentration on samples varies based on that geometric mean. Due to the absence of a control microRNA or a spiked miRNA, we consider that this methodology could be a good alternative, even though the number of samples used for this approach is limited. We have added new information in the limitation section of the manuscript.
“In addition, in this study, a targeted analysis of miRNAs was performed, so many potential correlations between breast milk miRNAs and maternal factors and diet could not be detected. One of the most common methods used to normalize gene expression data obtained by qPCR is the use of housekeeping genes. In the case of miRNAs it is highly complex to find housekeeping miRNAs. One of the options that can be used to normalize these data is to spike miRNAs into the samples. In this work we have chosen, because we did not used spiked miRNAs we opted to do a normalization of the data based on the geometric mean of all miRNAs. This is a technique more commonly employed in RNA-seq where there are thousands of miRNAs expressed. Although the method employed is not common in qPCR and has limitations such as not taking into account the individual variability of each sample, it may represent an alternative to compare the expression of a miRNA in a sample with respect to an geometrical mean of all miRNAs.”
Comment 3: Exosome isolation method does not give purified exosomes, but contaminated with many other proteins and molecules. Unless some additional purification is performed and verify by other methos such as electron microscopy, western blot or other, authors should empathize that exosomes are not pure in the limitation section. This has been already acknowledged.
Response 3: We have added the next comment in part of limitation section where we said that not verification was performed: “Therefore, these exosomes may be contaminated with proteins or other molecules”.
Comment 4: Figure 1 show relative abundance of miRNAs during lactation period. For example, miR-148a, relative abundance is lower <3 months and >12 months. They are in the negative scale. Shouldn’t they be in the positive scale and the other in the negative, so it indicates that their levels are higher (<3 months and >12 months). Is figure 1 (and all the others) correct? If so, please indicate in figure legend how to interpret those data. As it is now, it is misleading.
Response 4: Thank you very much for your comment. It is absolutely true that it can be confusing as it is expressed. Based on the formula used (Relative abundance= Cq target miRNA -Cq miRNA geometrical mean) and that the PCR cycle is inversely proportional to the amount of miRNA in the sample, a negative value indicates a higher initial load of microRNA in the sample because the Ct of the sample is lower than the geometric mean value of microRNAs. We have added the next information in the figure:
“The Cq value is inversely proportional to the initial amount of miRNAs in the sample. If the Cq value is lower than the geometric mean it indicates higher initial microRNA concentration. In these cases the result of the applied formula is negative. Therefore a higher negative value indicates a higher initial concentration of miRNA in the sample”.
Comment 5: Authors indicate that the aim of this work is to evaluate the effect of lactation time on the levels of miRNAs in breast milk and their correlation with milk composition (fatty acids, minerals, microbiota) and with dietary patterns of the mother. But no data on microbiota is presented. Please include data of microbiota.
Response 5: It was a mistake. Microbiota is not included in the analysis.
Comment 6: Authors use a previous published cohort of Spain mothers: Sanjulián L, Lamas A, Barreiro R, Cepeda A, Fente CA, Regal P. Bacterial Diversity of Breast Milk in Healthy Spanish Women: Evolution from Birth to Five Years Postpartum. Nutrients. 2021 Jul 14;13(7):2414. doi: 10.3390/nu13072414. PMID: 34371924; PMCID: PMC8308733. However, in this manuscript they use only 59 samples from a total of 99. Please indicate why not the 99 samples were used. They acknowledge that there is a low number of samples, but why not using the whole cohort of 99 samples (although they are still a low number of samples)
Response 6: Samples from this research project were used for various analyses. In this work we included the samples for which we had sufficient sample quantity in case we had to reanalyze the sample.
Comment 7: Authors do not show any results of the diet, minerals, fatty acids and dietary patterns. These data must be included both in material and methods section and in the results sections. These data are relevant and must be included in this manuscript.
Response 7: We have added in Material and Methods those analysis as well as the main results are presented in Tables in Results section.
Comment 8: Include the rationale to select those 11 miRNAs and not others.
Response 8: In the introduction section we have added the next information:
“The miRNAs were selected based on those that are highly expressed in breast milk, and for which at the time of study design there was evidence of their epigenetic role and their role in infant development and physiology [12,22,42,43].”
Comment 9: Authors indicate that the miRNA levels in breast milk after 12 months appear similar to those observed during the first months of lactation. This reviewer is not sure whether this affirmation is correct. With all the technical problems mentioned above, authors should have caution with the conclusions of this work.
Response 9: We have deleted that affirmation.
Comment 10: L66 modulating the reception of hormones? Please explain or rephrase
Response 10: It was rephrase to hormone receptors.
Reviewer 2 Report
Comments and Suggestions for Authors
Manuscript ID: foods-3517801
Type of manuscript: Article
Title: Influence of maternal diet and lactation time on the exosomal miRNA
cargo in breast milk.
Authors: Laura Sanjulián, Alexandre Lamas *, Rocío Barreiro, Alberto
Cepeda, Cristina Fente, Patricia Regal
General comment:
It is the objective of this interesting study to assess the presence of eleven dominant miRNAs (miR-148a-3p, miR-29b-3p, miR-125b-5p, miR-16 200c-3p, let-7f-5p, let-7b-5p, let-7a-5p, miR-92a, miR-181a-3p, miR-30a-5p, miR-155-5p) in breast milk exosomes and to determine the impact of lactation time as well as maternal diet on their expression levels. More information of the kinetics of milk exosomal miRs is important to understand milk miR-signaling and miR-dependent posttranslational epigenetic regulation. Especially, there is only limited information of milk exosomal miR expression after 6th months of lactation.
Unfortunately, the authors started sampling between 1st – 3rd months of lactation and missed miR expression changes occurring early at the transition from colostrum to transitional milk. The first and second postnatal weeks appear to be of most critical importance for systemic uptake of milk miRs due to a still immature intestinal permeability barrier allowing increased uptake of macromolecules and apparently of nano-sized milk exosomes.
Frazer LC, Good M. Intestinal epithelium in early life. Mucosal Immunol. 2022 Jun;15(6):1181-1187. doi: 10.1038/s41385-022-00579-8. Epub 2022 Nov 15. PMID: 36380094; PMCID: PMC10329854.
The authors missed the recent publication of Weil et al. providing experimental evidence for vertical transmission of milk miRs reaching the systemic circulation during neonatal time periods.
Weil PP, Reincke S, Hirsch CA, Giachero F, Aydin M, Scholz J, Jönsson F, Hagedorn C, Nguyen DN, Thymann T, Pembaur A, Orth V, Wünsche V, Jiang PP, Wirth S, Jenke ACW, Sangild PT, Kreppel F, Postberg J. Uncovering the gastrointestinal passage, intestinal epithelial cellular uptake, and AGO2 loading of milk miRNAs in neonates using xenomiRs as tracers. Am J Clin Nutr. 2023 Jun;117(6):1195-1210. doi: 10.1016/j.ajcnut.2023.03.016. Epub 2023 Mar 22. PMID: 36963568.
Specific comments:
Material and Methods:
The investigators did not screen for the expression of miR-30b-5p, which exhibits two base changes in the seed sequence compared to miR-30a-5p. It has been shown that eicosapentaenoic acid (EPA) via binding to free fatty receptor 4 (FFAR4) is able to upregulate the expression of miR-30b-5p, which is according to this manuscript not the matter for miR-30a-5p. MiR-30b-5p plays a key role in the development of brown adiposes tissue and thermogenesis, which is of importance for the newborn and especially for the preterm born infant.
Hu F, Wang M, Xiao T, Yin B, He L, Meng W, Dong M, Liu F. miR-30 promotes thermogenesis and the development of beige fat by targeting RIP140. Diabetes. 2015 Jun;64(6):2056-68. doi: 10.2337/db14-1117. Epub 2015 Jan 9. PMID: 25576051; PMCID: PMC4876748.
Kim J, Okla M, Erickson A, Carr T, Natarajan SK, Chung S. Eicosapentaenoic Acid Potentiates Brown Thermogenesis through FFAR4-dependent Up-regulation of miR-30b and miR-378. J Biol Chem. 2016 Sep 23;291(39):20551-62. doi: 10.1074/jbc.M116.721480. Epub 2016 Aug 3. PMID: 27489163; PMCID: PMC5034049.
Gharanei S, Shabir K, Brown JE, Weickert MO, Barber TM, Kyrou I, Randeva HS. Regulatory microRNAs in Brown, Brite and White Adipose Tissue. Cells. 2020 Nov 16;9(11):2489. doi: 10.3390/cells9112489. PMID: 33207733; PMCID: PMC7696849.
Results:
The authors should present a comparison of miR-148a-3p and miR-30a-5p expression between mothers that delivered naturally versus C-section as oxytocin stimulates the expression of both miRs.
Gutman-Ido E, Reif S, Musseri M, Schabes T, Golan-Gerstl R. Oxytocin Regulates the Expression of Selected Colostrum-derived microRNAs. J Pediatr Gastroenterol Nutr. 2022 Jan 1;74(1):e8-e15. doi: 10.1097/MPG.0000000000003277. PMID: 34371509.
The detection of miR-30a-5p in the 1-3 months group may be the result of oxytocin-stimulated miR expression by vaginal birth.
Furthermore, both miR-148a-3p and miR-125-5p expression are reduced in human breast milk of mothers who delivered via cesarean section compared to vaginal delivery.
Chiba T, Kooka A, Kowatari K, Yoshizawa M, Chiba N, Takaguri A, Fukushi Y, Hongo F, Sato H, Wada S. Expression profiles of hsa-miR-148a-3p and hsa-miR-125b-5p in human breast milk and infant formulae. Int Breastfeed J. 2022 Jan 3;17(1):1. doi: 10.1186/s13006-021-00436-7. PMID: 34980190; PMCID: PMC8725387.
Both, miRs target TP53 mRNA and thus have an impact on a vast regulatory network of p53-regulated genes like IGF-1 receptor.
Neuberg M, Buckbinder L, Seizinger B, Kley N. The p53/IGF-1 receptor axis in the regulation of programmed cell death. Endocrine. 1997 Aug;7(1):107-9. doi: 10.1007/BF02778075. PMID: 9449044.
Discussion
Line 257
Function of miR-148a-3p targeting DNMT1
MiR-148a-3p not only exerts anti-inflammatory functions but via targeting DNMT1 mRNA increases the expression of FoxP3, the master transcription factor of regulatory T cells (Tregs), enhancing their expansion for the control of intestinal immune tolerance and eventually systemic immune tolerance development.
Polansky JK, Kretschmer K, Freyer J, Floess S, Garbe A, Baron U, Olek S, Hamann A, von Boehmer H, Huehn J. DNA methylation controls Foxp3 gene expression. Eur J Immunol. 2008 Jun;38(6):1654-63. doi: 10.1002/eji.200838105. PMID: 18493985.
Lal G, Bromberg JS. Epigenetic mechanisms of regulation of Foxp3 expression. Blood. 2009 Oct 29;114(18):3727-35. doi: 10.1182/blood-2009-05-219584. Epub 2009 Jul 29. PMID: 19641188; PMCID: PMC2773485.
Melnik BC, John SM, Schmitz G. Milk: an exosomal microRNA transmitter promoting thymic regulatory T cell maturation preventing the development of atopy? J Transl Med. 2014 Feb 12;12:43. doi: 10.1186/1479-5876-12-43. PMID: 24521175; PMCID: PMC3930015.
Polansky JK, Schreiber L, Thelemann C, Ludwig L, Krüger M, Baumgrass R, Cording S, Floess S, Hamann A, Huehn J. Methylation matters: binding of Ets-1 to the demethylated Foxp3 gene contributes to the stabilization of Foxp3 expression in regulatory T cells. J Mol Med (Berl). 2010 Oct;88(10):1029-40. doi: 10.1007/s00109-010-0642-1. Epub 2010 Jun 24. PMID: 20574810; PMCID: PMC2943068.
Melnik BC, Stremmel W, Weiskirchen R, John SM, Schmitz G. Exosome-Derived MicroRNAs of Human Milk and Their Effects on Infant Health and Development. Biomolecules. 2021 Jun 7;11(6):851. doi: 10.3390/biom11060851. PMID: 34200323; PMCID: PMC8228670.
Line 295
miR-148a-3p directly targets TP53 mRNA.
Guo MM, Zhang K, Zhang JH. Human Breast Milk-Derived Exosomal miR-148a-3p Protects Against Necrotizing Enterocolitis by Regulating p53 and Sirtuin 1. Inflammation. 2022 Jun;45(3):1254-1268. doi: 10.1007/s10753-021-01618-5. Epub 2022 Jan 29. PMID: 35091894.
Line 315
When discussing the data of Hicks et al. (2022), it should be noted that Hicks and coworkers presented data on complete human milk and NOT on milk exosomes. The milk lipid fraction (Milk fat granules) contains abundant miRs including miR-148a-3p.
Munch EM, Harris RA, Mohammad M, Benham AL, Pejerrey SM, Showalter L, Hu M, Shope CD, Maningat PD, Gunaratne PH, Haymond M, Aagaard K. Transcriptome profiling of microRNA by Next-Gen deep sequencing reveals known and novel miRNA species in the lipid fraction of human breast milk. PLoS One. 2013;8(2):e50564. doi: 10.1371/journal.pone.0050564. Epub 2013 Feb 13. PMID: 23418415; PMCID: PMC3572105.
Thus, the value of comparing own exosome-related results with the Hicks paper (complete milk) is limited.
Line 360
Discussion of milk miR uptake into systemic circulation
It should be noted that cow milk consumption-associated bovine milk miR uptake in adult human volunteers is not comparable to the postnatal period characterized by higher intestinal permeability of newborn infants. The evidence of Weil et al. should be considered in this context.
Weil PP, Reincke S, Hirsch CA, Giachero F, Aydin M, Scholz J, Jönsson F, Hagedorn C, Nguyen DN, Thymann T, Pembaur A, Orth V, Wünsche V, Jiang PP, Wirth S, Jenke ACW, Sangild PT, Kreppel F, Postberg J. Uncovering the gastrointestinal passage, intestinal epithelial cellular uptake, and AGO2 loading of milk miRNAs in neonates using xenomiRs as tracers. Am J Clin Nutr. 2023 Jun;117(6):1195-1210. doi: 10.1016/j.ajcnut.2023.03.016. Epub 2023 Mar 22. PMID: 36963568.
Conclusions
The reviewer is not convinced that addition of miRs to formula makes artificial systems more adapted and similar to breast milk. This stays elusive and the sentence
„Knowing the miRNA profiles at each stage of lactation may be relevant in the future for the development of infant formulas that are more adapted and similar to breast milk.”
should be deleted.
Author Response
Comment 1:
It is the objective of this interesting study to assess the presence of eleven dominant miRNAs (miR-148a-3p, miR-29b-3p, miR-125b-5p, miR-16 200c-3p, let-7f-5p, let-7b-5p, let-7a-5p, miR-92a, miR-181a-3p, miR-30a-5p, miR-155-5p) in breast milk exosomes and to determine the impact of lactation time as well as maternal diet on their expression levels. More information of the kinetics of milk exosomal miRs is important to understand milk miR-signaling and miR-dependent posttranslational epigenetic regulation. Especially, there is only limited information of milk exosomal miR expression after 6th months of lactation.
Unfortunately, the authors started sampling between 1st – 3rd months of lactation and missed miR expression changes occurring early at the transition from colostrum to transitional milk. The first and second postnatal weeks appear to be of most critical importance for systemic uptake of milk miRs due to a still immature intestinal permeability barrier allowing increased uptake of macromolecules and apparently of nano-sized milk exosomes.
Frazer LC, Good M. Intestinal epithelium in early life. Mucosal Immunol. 2022 Jun;15(6):1181-1187. doi: 10.1038/s41385-022-00579-8. Epub 2022 Nov 15. PMID: 36380094; PMCID: PMC10329854.
Response 1: We fully agree with the reviewer. Unfortunately for this study we did not have such samples. It is something we are trying to collect in a new study we are going to start. We have added in the introduction section a couple of lines regarding the importance of a inmature intestinal permeability barrier in the first period of life.
Comment 2:
The authors missed the recent publication of Weil et al. providing experimental evidence for vertical transmission of milk miRs reaching the systemic circulation during neonatal time periods.
Weil PP, Reincke S, Hirsch CA, Giachero F, Aydin M, Scholz J, Jönsson F, Hagedorn C, Nguyen DN, Thymann T, Pembaur A, Orth V, Wünsche V, Jiang PP, Wirth S, Jenke ACW, Sangild PT, Kreppel F, Postberg J. Uncovering the gastrointestinal passage, intestinal epithelial cellular uptake, and AGO2 loading of milk miRNAs in neonates using xenomiRs as tracers. Am J Clin Nutr. 2023 Jun;117(6):1195-1210. doi: 10.1016/j.ajcnut.2023.03.016. Epub 2023 Mar 22. PMID: 36963568.
Response 2: The refence was included in the discussion section.
Specific comments:
Material and Methods:
Comment 3:
The investigators did not screen for the expression of miR-30b-5p, which exhibits two base changes in the seed sequence compared to miR-30a-5p. It has been shown that eicosapentaenoic acid (EPA) via binding to free fatty receptor 4 (FFAR4) is able to upregulate the expression of miR-30b-5p, which is according to this manuscript not the matter for miR-30a-5p. MiR-30b-5p plays a key role in the development of brown adiposes tissue and thermogenesis, which is of importance for the newborn and especially for the preterm born infant.
Hu F, Wang M, Xiao T, Yin B, He L, Meng W, Dong M, Liu F. miR-30 promotes thermogenesis and the development of beige fat by targeting RIP140. Diabetes. 2015 Jun;64(6):2056-68. doi: 10.2337/db14-1117. Epub 2015 Jan 9. PMID: 25576051; PMCID: PMC4876748.
Kim J, Okla M, Erickson A, Carr T, Natarajan SK, Chung S. Eicosapentaenoic Acid Potentiates Brown Thermogenesis through FFAR4-dependent Up-regulation of miR-30b and miR-378. J Biol Chem. 2016 Sep 23;291(39):20551-62. doi: 10.1074/jbc.M116.721480. Epub 2016 Aug 3. PMID: 27489163; PMCID: PMC5034049.
Gharanei S, Shabir K, Brown JE, Weickert MO, Barber TM, Kyrou I, Randeva HS. Regulatory microRNAs in Brown, Brite and White Adipose Tissue. Cells. 2020 Nov 16;9(11):2489. doi: 10.3390/cells9112489. PMID: 33207733; PMCID: PMC7696849.
Response 3: Many thanks for the comment. We have added some of this information in discussion section.
Results:
Comment 4:
The authors should present a comparison of miR-148a-3p and miR-30a-5p expression between mothers that delivered naturally versus C-section as oxytocin stimulates the expression of both miRs.
Gutman-Ido E, Reif S, Musseri M, Schabes T, Golan-Gerstl R. Oxytocin Regulates the Expression of Selected Colostrum-derived microRNAs. J Pediatr Gastroenterol Nutr. 2022 Jan 1;74(1):e8-e15. doi: 10.1097/MPG.0000000000003277. PMID: 34371509.
The detection of miR-30a-5p in the 1-3 months group may be the result of oxytocin-stimulated miR expression by vaginal birth.
Furthermore, both miR-148a-3p and miR-125-5p expression are reduced in human breast milk of mothers who delivered via cesarean section compared to vaginal delivery.
Chiba T, Kooka A, Kowatari K, Yoshizawa M, Chiba N, Takaguri A, Fukushi Y, Hongo F, Sato H, Wada S. Expression profiles of hsa-miR-148a-3p and hsa-miR-125b-5p in human breast milk and infant formulae. Int Breastfeed J. 2022 Jan 3;17(1):1. doi: 10.1186/s13006-021-00436-7. PMID: 34980190; PMCID: PMC8725387.
Both, miRs target TP53 mRNA and thus have an impact on a vast regulatory network of p53-regulated genes like IGF-1 receptor.
Neuberg M, Buckbinder L, Seizinger B, Kley N. The p53/IGF-1 receptor axis in the regulation of programmed cell death. Endocrine. 1997 Aug;7(1):107-9. doi: 10.1007/BF02778075. PMID: 9449044.
Response 4: We really appreciate the comments of the reviewer. In this study we didn´t find significant differences between mother with vaginal delivery and C-section.
Discussion
Comment 5
Line 257
Function of miR-148a-3p targeting DNMT1
MiR-148a-3p not only exerts anti-inflammatory functions but via targeting DNMT1 mRNA increases the expression of FoxP3, the master transcription factor of regulatory T cells (Tregs), enhancing their expansion for the control of intestinal immune tolerance and eventually systemic immune tolerance development.
Polansky JK, Kretschmer K, Freyer J, Floess S, Garbe A, Baron U, Olek S, Hamann A, von Boehmer H, Huehn J. DNA methylation controls Foxp3 gene expression. Eur J Immunol. 2008 Jun;38(6):1654-63. doi: 10.1002/eji.200838105. PMID: 18493985.
Lal G, Bromberg JS. Epigenetic mechanisms of regulation of Foxp3 expression. Blood. 2009 Oct 29;114(18):3727-35. doi: 10.1182/blood-2009-05-219584. Epub 2009 Jul 29. PMID: 19641188; PMCID: PMC2773485.
Melnik BC, John SM, Schmitz G. Milk: an exosomal microRNA transmitter promoting thymic regulatory T cell maturation preventing the development of atopy? J Transl Med. 2014 Feb 12;12:43. doi: 10.1186/1479-5876-12-43. PMID: 24521175; PMCID: PMC3930015.
Polansky JK, Schreiber L, Thelemann C, Ludwig L, Krüger M, Baumgrass R, Cording S, Floess S, Hamann A, Huehn J. Methylation matters: binding of Ets-1 to the demethylated Foxp3 gene contributes to the stabilization of Foxp3 expression in regulatory T cells. J Mol Med (Berl). 2010 Oct;88(10):1029-40. doi: 10.1007/s00109-010-0642-1. Epub 2010 Jun 24. PMID: 20574810; PMCID: PMC2943068.
Melnik BC, Stremmel W, Weiskirchen R, John SM, Schmitz G. Exosome-Derived MicroRNAs of Human Milk and Their Effects on Infant Health and Development. Biomolecules. 2021 Jun 7;11(6):851. doi: 10.3390/biom11060851. PMID: 34200323; PMCID: PMC8228670.
Response 5: We have added this information in the discussion section.
Comment 6:
Line 295
miR-148a-3p directly targets TP53 mRNA.
Guo MM, Zhang K, Zhang JH. Human Breast Milk-Derived Exosomal miR-148a-3p Protects Against Necrotizing Enterocolitis by Regulating p53 and Sirtuin 1. Inflammation. 2022 Jun;45(3):1254-1268. doi: 10.1007/s10753-021-01618-5. Epub 2022 Jan 29. PMID: 35091894.
Response 6: The text was modified accordingly.
Comment 7:
Line 315
When discussing the data of Hicks et al. (2022), it should be noted that Hicks and coworkers presented data on complete human milk and NOT on milk exosomes. The milk lipid fraction (Milk fat granules) contains abundant miRs including miR-148a-3p.
Munch EM, Harris RA, Mohammad M, Benham AL, Pejerrey SM, Showalter L, Hu M, Shope CD, Maningat PD, Gunaratne PH, Haymond M, Aagaard K. Transcriptome profiling of microRNA by Next-Gen deep sequencing reveals known and novel miRNA species in the lipid fraction of human breast milk. PLoS One. 2013;8(2):e50564. doi: 10.1371/journal.pone.0050564. Epub 2013 Feb 13. PMID: 23418415; PMCID: PMC3572105.
Thus, the value of comparing own exosome-related results with the Hicks paper (complete milk) is limited.
Response 7: Absolutely correct, this is an important detail. In the text we indicate that the results of Hick et al. are in the lipid fraction.
Comment 8
Line 360
Discussion of milk miR uptake into systemic circulation
It should be noted that cow milk consumption-associated bovine milk miR uptake in adult human volunteers is not comparable to the postnatal period characterized by higher intestinal permeability of newborn infants. The evidence of Weil et al. should be considered in this context.
Weil PP, Reincke S, Hirsch CA, Giachero F, Aydin M, Scholz J, Jönsson F, Hagedorn C, Nguyen DN, Thymann T, Pembaur A, Orth V, Wünsche V, Jiang PP, Wirth S, Jenke ACW, Sangild PT, Kreppel F, Postberg J. Uncovering the gastrointestinal passage, intestinal epithelial cellular uptake, and AGO2 loading of milk miRNAs in neonates using xenomiRs as tracers. Am J Clin Nutr. 2023 Jun;117(6):1195-1210. doi: 10.1016/j.ajcnut.2023.03.016. Epub 2023 Mar 22. PMID: 36963568.
Response 8: We have added this information to the discussion section.
Conclusions
Comment 9
The reviewer is not convinced that addition of miRs to formula makes artificial systems more adapted and similar to breast milk. This stays elusive and the sentence
„Knowing the miRNA profiles at each stage of lactation may be relevant in the future for the development of infant formulas that are more adapted and similar to breast milk.”
should be deleted.
Response 9: We have deleted this sentence.
Reviewer 3 Report
Comments and Suggestions for Authors
This manuscript presents an intriguing study on the effects of maternal diet and lactation period on exosomal miRNA transport in human milk. The study is well-conducted, and the results provide valuable insights into the dynamic changes of miRNA profiles during lactation. The strengths of this research include comprehensive analysis of multiple miRNAs and their correlations with milk composition and maternal dietary patterns. However, there are several areas requiring attention, particularly regarding sample size, statistical analysis, and discussion of limitations.
Comments to Authors:
The introduction is well-written and provides clear rationale for the study. Authors effectively highlight the importance of breastfeeding and the potential roles of miRNAs in infant development. Consider adding a brief section explaining potential mechanisms through which maternal diet might influence miRNA profiles in breast milk to better contextualize the study.
The methodology is described in sufficient detail to enable reproducibility. The use of a cross-sectional study design and a diverse cohort of 59 mothers is appropriate. However, the sample size is relatively small, particularly in the extended lactation group (16 participants). This may limit the statistical power and generalizability of the findings. Authors should discuss this limitation more thoroughly. Consider performing a power analysis to justify sample size adequacy or discuss potential Type II errors arising from limited sample size.
Results are clearly presented with detailed statistics and figures. The findings on lactation-dependent changes in miRNA levels are significant and add to existing literature. However, the high Ct values for several miRNAs resulted in undetectable levels in some samples, which may affect result reliability. Authors should clarify how this potential bias was mitigated. Additional details about normalization methods for qPCR data are recommended. The use of geometric mean of all Cq values as an endogenous control is unconventional. Justify this choice and discuss its potential impact on results.
The discussion is comprehensive and contextualizes findings within prior research. Authors effectively emphasize the implications of their results for infant development and maternal nutrition. However, the discussion of limitations is somewhat brief. Expand on how small sample size and high Ct values may influence study conclusions. Consider discussing potential mechanisms through which maternal diet and lactation period might modulate miRNA profiles, based on observed correlations. This could provide deeper insights into the findings.
The conclusions are well-supported by data and highlight the dynamic nature of miRNA profiles in breast milk. Authors appropriately propose future research directions. Consider emphasizing potential clinical applications of the findings, such as developing infant formulas that mimic breast milk miRNA profiles.
Author Response
Comment 1: The introduction is well-written and provides clear rationale for the study. Authors effectively highlight the importance of breastfeeding and the potential roles of miRNAs in infant development. Consider adding a brief section explaining potential mechanisms through which maternal diet might influence miRNA profiles in breast milk to better contextualize the study.
Response 1: We have added the next in the introduction section:
“A previous study found correlation between diets rich in animal protein and diets rich plant protein and breast milk miRNAs profiles [36]. For example, dietary ingestion of polyphenols was reflected in the profile of breas milk miRNAs. In the same way, the type of diet has an influence in gut microbiota, which may also be reflected in the miRNAs present breast milk [36]. These modifications may have an impact on the development of the infant.”
Comment 2: The methodology is described in sufficient detail to enable reproducibility. The use of a cross-sectional study design and a diverse cohort of 59 mothers is appropriate. However, the sample size is relatively small, particularly in the extended lactation group (16 participants). This may limit the statistical power and generalizability of the findings. Authors should discuss this limitation more thoroughly. Consider performing a power analysis to justify sample size adequacy or discuss potential Type II errors arising from limited sample size.
Response 2: In the limitation section of the manuscript we have added the next information: “The number of samples included is low and it was not a longitudinal study of the volunteers but rather a cross-sectional observational study. This limits the comparison between different lactation periods. It is important to mention that the extended lactation group only has 16 volunteers with a very wide range of lactation time, and therefore, the conclusions drawn are limited. The limited number of samples in each group reduces the statistical power of this study, which may result in some differences or correlations between microRNAs and diet and maternal factors not being detected. Likewise, the non-detection of some miRNAs in various samples also has an impact on the statistical results of this study.”
Comment 3: Results are clearly presented with detailed statistics and figures. The findings on lactation-dependent changes in miRNA levels are significant and add to existing literature. However, the high Ct values for several miRNAs resulted in undetectable levels in some samples, which may affect result reliability. Authors should clarify how this potential bias was mitigated. Additional details about normalization methods for qPCR data are recommended. The use of geometric mean of all Cq values as an endogenous control is unconventional. Justify this choice and discuss its potential impact on results.
Response 3: For Cq normalization. In Figure 1 we added information in the legend of the figure on how the data is interpreted for a better understanding. In limitation section we discuss about the main normalization strategies and the reason to choose this normalization strategy in this study.
“In addition, in this study, a targeted analysis of miRNAs was performed, so many potential correlations between breast milk miRNAs and maternal factors and diet could not be detected. One of the most common methods used to normalize gene expression data obtained by qPCR is the use of housekeeping genes. In the case of miRNAs it is highly complex to find housekeeping miRNAs. One of the options that can be used to normalize these data is to spike miRNAs into the samples. In this work we have chosen, because we did not used spiked miRNAs we opted to do a normalization of the data based on the geometric mean of all miRNAs. This is a technique more commonly employed in RNA-seq where there are thousands of miRNAs expressed. Although the method employed is not common in qPCR and has limitations such as not taking into account the individual variability of each sample, it may represent an alternative to compare the expression of a miRNA in a sample with respect to an geometrical mean of all miRNAs.”
Comment 4: The discussion is comprehensive and contextualizes findings within prior research. Authors effectively emphasize the implications of their results for infant development and maternal nutrition. However, the discussion of limitations is somewhat brief. Expand on how small sample size and high Ct values may influence study conclusions. Consider discussing potential mechanisms through which maternal diet and lactation period might modulate miRNA profiles, based on observed correlations. This could provide deeper insights into the findings.
Response 4: In the limitation section we have added some information about the limitations related to sample size:
“This limits the comparison between different lactation periods. It is important to mention that the extended lactation group only has 16 volunteers with a very wide range of lactation time, and therefore, the conclusions drawn are limited. The limited number of samples in each group reduces the statistical power of this study, which may result in some differences or correlations between microRNAs and diet and maternal factors not being detected. A lower number of samples per group could increase the variability and influence Anova and correlation analysis”.
We also added:
“. In addition, in this study, a targeted analysis of miRNAs was performed, so many potential correlations between breast milk miRNAs and maternal factors and diet could not be detected.”
For diet and miRNAs, in discussion section we added:
“Fruit is a source of polyphenols and previous studies have observed a correlation between breast milk microRNAs and these bioactive molecules [36]. In future studies, it will be interesting to determine whether it is the polyphenols themselves that directly influence miRNA expression in cells or whether it is the diet associated with polyphenol consumption that actually influences miRNA levels”
“Likewise, in a previous study, differences in miRNAs in breast milk related to cell proliferation have been observed between diets rich in vegetable protein and diets rich in animal protein, being overexpressed in the latter [36].”
Comment 5: The conclusions are well-supported by data and highlight the dynamic nature of miRNA profiles in breast milk. Authors appropriately propose future research directions. Consider emphasizing potential clinical applications of the findings, such as developing infant formulas that mimic breast milk miRNA profiles
Response 5: Many thanks for the comments. They were very constructive
Round 2
Reviewer 1 Report
Comments and Suggestions for Authors
Authors must remove the wording "Curiously, extended lactation milk samples tend to resemble 1-2 month samples" from the abstract as based on the results and all the technical problems in the analysis it can not be acertain.
Please check the references carefully as some numbers appear to have been mixed up in the new version of the manuscript.
L58-61. Although miRNAs may exhibit increased resistance to degradation, this does not necessarily mean they are resistant at all. If the authors carefully review the literature, they will notice that there is only a small increase in resistance. It is not certain whether this is sufficient to exert a biological effect, as there are authors for and against. Note that there is a difficult path to potential effects. The authors may review and use this review to gain further insight (PMID: 29627443).
L180 seems to be reference 33, please verify and include it.
L492-496 there are additional information that suggest that miRNAs can be modualted by fatty acids, PMID: 25671565 which can be included in the manuscript.
Author Response
Comment 1: Authors must remove the wording "Curiously, extended lactation milk samples tend to resemble 1-2 month samples" from the abstract as based on the results and all the technical problems in the analysis it can not be acertain.
Response 1: The sentence was deleted.
Comment 2: Please check the references carefully as some numbers appear to have been mixed up in the new version of the manuscript.
Response 2: Revised.
Comment 3: L58-61. Although miRNAs may exhibit increased resistance to degradation, this does not necessarily mean they are resistant at all. If the authors carefully review the literature, they will notice that there is only a small increase in resistance. It is not certain whether this is sufficient to exert a biological effect, as there are authors for and against. Note that there is a difficult path to potential effects. The authors may review and use this review to gain further insight (PMID: 29627443).
Response 3: We have modified the sentence to indicate that could increase the resistance. We also highlighted that the evidence on the biological effect is still inconclusive.
Comment 4: L180 seems to be reference 33, please verify and include it.
Response 4: Reference added.
L492-496 there are additional information that suggest that miRNAs can be modualted by fatty acids, PMID: 25671565 which can be included in the manuscript.
Response 5: This information has been added to the discussion.
Reviewer 3 Report
Comments and Suggestions for Authors
The article has been significantly improved. It is recommended to accept the article.
Author Response
Comment 1: The article has been significantly improved. It is recommended to accept the article.
Response: We really appreciate the reviewer effort to revise this manuscript.